# Ultrasonographic characteristics of major salivary glands in anti-centromere antibody-positive primary Sjögren's syndrome

Hong Ki Min[1], Se-Hee Kim[1], Youngjae Park[2], Kyung-Ann Lee[3], Seung-Ki Kwok[2], Sang-Heon Lee[4], Hae-Rim Kim[4]*

1 Division of Rheumatology, Department of Internal Medicine, Konkuk University Medical Center, Seoul, Republic of Korea, 2 Division of Rheumatology, Department of Internal Medicine, Seoul St. Mary's Hospital, College of Medicine, The Catholic University of Korea, Seoul, Republic of Korea, 3 Division of Rheumatology, Department of Internal Medicine, Soonchunhyang University Seoul hospital, Seoul, Republic of Korea, 4 Division of Rheumatology, Department of Internal Medicine, Research Institute of Medical Science, Konkuk University Medical Center, Konkuk University School of Medicine, Seoul, Republic of Korea

* kimhaerim@kuh.ac.kr

**Data Availability Statement:** All relevant data are within the manuscript and its Supporting Information files.

## Abstract

### Purpose

To investigate salivary gland ultrasonography (SGUS) findings in primary Sjögren's syndrome (pSS) patients positive for the anti-centromere antibody (ACA) and compare these with those in ACA-negative pSS patients.

### Methods

We analyzed demographic, clinical, laboratory, and SGUS data of pSS patients who fulfilled the 2002 American-European Consensus Group classification criteria for pSS. SGUS findings of four major salivary glands (bilateral parotid and submandibular glands) were scored in five categories and compared between ACA-positive and ACA-negative pSS patients. Linear regression analysis was performed to elucidate the factors associated with SGUS score.

### Results

In total, 121 pSS patients were enrolled (19, ACA-positive). The ACA-positive patients were older (67.0 vs 58.0 years, $P = 0.028$), whereas anti-Ro/SSA and anti-La/SSB positivity was more prevalent in the ACA-negative group (89.2% vs 21.1%, $P < 0.001$, and 47.1% vs 10.5%, $P = 0.007$, respectively). The total SGUS and hypoechoic area scores were lower in ACA-positive patients (16.0 vs 23.0, $P = 0.027$, and 4.0 vs 7.0, $P = 0.004$, respectively). In univariate regression analysis, being positive for unstimulated salivary flow rate (USFR < 1.5 ml/15 min), anti-Ro/SSA, and rheumatoid factor were positively associated whereas ACA positivity was negatively associated with the SGUS score. In multivariate regression analysis, being positive for USFR, anti-Ro/SSA, and rheumatoid factor showed significant association with the SGUS score.

**Funding:** This research was supported by a grant from the Basic Science Research Program through the National Research Foundation of Korea (NRF) funded by the Ministry of Education, Science and Technology, Republic of Korea (NRF-2018R1D1A1A02050982, Receiver: HR Kim). The funders had no role in study design, data collection and analysis, decision to publish, or preparation of the manuscript.

**Competing interests:** The authors declare that they have no competing interests relevant to this manuscript.

## Conclusions

ACA-positive pSS patients showed a lower SGUS score than ACA-negative patients, which was especially prominent in the hypoechoic area component.

## Introduction

Primary Sjögren's syndrome (pSS) is a systemic autoimmune disease which primarily affects exocrine glands and causes sicca symptoms. The prevalence of pSS varies between 0.01–0.72%, with the condition occurring about 10 times more frequently in females than in males [1]. The pathogenesis of pSS is complex and even dysbiosis is associated with pSS pathogenesis [2]. The manifestations of pSS are not limited to sicca symptoms but may also be accompanied by arthralgia, fatigue, cutaneous symptoms, pulmonary involvement, cystitis, and even sleep disturbance [3–6]. A diagnosis of pSS can be made by quantifying the degree of xerostomia/xerophthalmia, a special stain for the eyes, from histologic findings of minor salivary glands, and the presence of pSS-associated autoantibodies such as anti-Ro/SSA and anti-La/SSB [7,8].

Major salivary gland ultrasonography (SGUS) has been suggested as an adjuvant tool for pSS diagnosis and shows a similar diagnostic power to sialoscintigraphy or minor salivary gland biopsy [9]. Furthermore, Cornec et al has shown that using a SGUS score with the 2002 American-European Consensus Group (AECG) classification criteria enhances diagnostic power [10]. Since SGUS has the advantage of finding structural changes in the salivary gland of pSS patients, it has been suggested as an early diagnostic tool [11]. The SGUS score correlates with unstimulated whole saliva flow rate (USFR) and the focus score of minor salivary gland biopsy [11,12]. A recent study has shown that hyperechoic foci found during SGUS also correlates with USFR [13], which suggests that SGUS findings could predict not only the structural changes in salivary glands, but also the functional and histologic changes. Furthermore, the SGUS showed significant association with disease activity measured by European League Against Rheumatism Sjögren's syndrome disease activity index, which imply that SGUS is not only useful for diagnosis of pSS but also for predicting clinical activity of pSS [14].

The two autoantibodies, anti-Ro/SSA and anti-La/SSB, are known to be specific for pSS. These antibodies directly combine with ribonucleoprotein complexes, which are composed of five proteins; Ro 52 kDa, Ro 60 kDa, La, calreticulin, and nucleolin [15]. Although anti-Ro/SSA and anti-La/SSB antibodies are found in about 80% of pSS patients, several other autoantibodies are also present [16]. Among patients with these atypical autoantibodies, anti-centromere antibody (ACA)-positive patients are perceived to be a unique pSS subgroup [15,17]. However, the SGUS findings of ACA-positive pSS patients have not yet been evaluated.

We primarily aimed to evaluate the SGUS findings of ACA-positive pSS patients and compare these with ACA-negative pSS patients. Furthermore, we evaluated the influence of autoantibodies, including ACA, on the SGUS score.

## Materials and methods

### Study population

All patients who visited the rheumatology clinic of single tertiary hospital for xerostomia/xerophthalmia or known pSS from June 2016 to April 2020 underwent SGUS evaluation. Any patient who fulfilled the 2002 AECG classification criteria for pSS was included in the analysis [18]. Patients with other autoimmune diseases, such as systemic lupus erythematosus, systemic

sclerosis, inflammatory myositis, rheumatoid arthritis, and mixed connective tissue disease, were excluded. In addition, patients with malignancies were excluded. The present study was conducted in accordance with the Declaration of Helsinki and Good Clinical Practice guidelines and was approved by the Institutional Review Board (IRB) of Konkuk University Medical Center (IRB number: 2020-05-045). The requirement for written informed consent was waived by the IRB of Konkuk University Medical Center because data were collected retrospectively.

Demographic, laboratory, and clinical data were collected. Exocrine gland function (USFR and Schirmer I test) and laboratory tests were performed at the same time as SGUS. The USFR was measured for 15 minutes, with less than 1.5 ml of saliva observed in this time deemed as a positive finding. A positive result for the Schirmer I test was defined as less than 5 mm of wetting in filter paper after 5 minutes [8]. Indirect immunofluorescence assays using HEp-2 cells were performed to detect the anti-nuclear antibody (ANA), and an ANA titer of over 1:320 was considered as positive, in accordance with the 2012 ACR classification criteria [7]. Anti-Ro/SSA, anti-La/SSB, and ACA were measured by enzyme-linked immunosorbent assay. Rheumatoid factor (RF) titer was measured via the nephelometry method, and a normal reference range was established as less than 18 IU/ml. Hypergammaglobulinemia was defined as an immunoglobulin G level over 1600 mg/L, according to the biological domain of the EULAR Sjögren's syndrome disease activity index [19]. Reference ranges for complement 3 and 4 were 86 to 160, and 17 to 47 mg/dL, respectively.

## Major salivary gland ultrasonography assessment

We performed SGUS in the bilateral parotid and submandibular glands and scored the findings semi-quantitatively according to Hocevar et al [20]. The patients were placed on bed by supine position, and SGUS for parotid gland was assessed after patient's head was maximally tilted to opposite side, then scanned in the retromandibular fossa by longitudinal and transverse planes. After assessing parotid gland, then head was maximally tilted to backward to assess submandibular gland in posterior part of the submandibular triangle by longitudinal planes. The thyroid gland was also scanned to compare the echogenicity of salivary gland. A semi-quantitative SGUS score was determined by grading in 5 categories: echogenicity, homogeneity, hypoechoic areas, hyperechoic foci, and clearness of salivary gland borders. The score for echogenicity was 0 when the salivary gland parenchyma showed similar echogenicity to the thyroid gland and 1 when echogenicity of the salivary gland was decreased. Homogeneity was scored from 0 to 3 (grade 0 for a homogeneous gland, 1 for mild inhomogeneity, 2 for definite inhomogeneity, and 3 for a grossly inhomogeneous gland). The hypoechoic areas were similarly graded (grade 0 for the absence of hypoechoic lesions, 1 for a few hypoechoic lesions, 2 for several, and 3 for diffuse numerous hypoechoic lesions). Hyperechoic foci were graded from 0 to 3 for the parotid gland (grade 0 for the absence of hyperechoic foci, 1 for a few, 2 for several, and 3 for diffuse numerous hyperechoic foci) and from 0 to 1 for the submandibular gland (grade 0 for the absence and 1 for the presence of hyperechoic foci). Finally, the clearance of the gland border was also graded from 0 to 3 (grade 0 for a clear border, 1 for a partially less delineated border, 2 for an ill-defined delineated border, and 3 for a non-visible border). The total score for each parotid and submandibular gland ranged from 0 to 13 and 0 to 11, respectively. The total possible SGUS score was 48, and a score over 14 was defined as compatible with pSS [21]. Parenchymal power Doppler ultrasonography (PDUS) was also graded from 0 to 3 as follows: grade 0 for no abnormal flow; 1 for up to three spots of PD signals, up to two confluent spots, or one confluent spot plus up to two single spots; 2 for PD signals in less than half of the gland parenchyma ($\leq$ 50%); and 3 for PD signals in more than half of the gland

parenchyma ($> 50\%$) [21]. The PDUS score for each parotid and submandibular gland ranged 0 to 3, and the total PDUS score could reach 12. An experienced rheumatologist (K.A. Lee) performed SGUS and PDUS and these were scored by two independent rheumatologist (H.K. Min and S.H. Kim) for twice, who was blinded for the patient's information. All ultrasonography examinations were conducted using an HD15 PureWave US system (Philips Ultrasound, Bothell, WA, USA) device with a 5–12 MHz multi-frequency linear probe.

### Sample size calculation

Primary endpoint for present study was comparing total SGUS score between ACA positive and ACA negative patients with pSS. In previous study [21], difference of SGUS score between pSS and idiopathic sicca syndrome groups was 17 with standard deviation (SD) 13. On the basis of $\alpha$-error 5%, statistic power 90%, and drop-out rate 10%, at least 14 patients were required for each group. The present study was conducted as retrospective manner, we included as many patients as possible.

### Propensity score matching

As baseline characteristics of ACA positive and negative patients with pSS showed in aspect of age and USFR, we performed propensity score (PS) matched ACA negative pSS group by using nearest-neighboring with 1:2 ratio. Age and USFR were imputed as variables in PS matching. The SGUS finding between ACA positive and PS-matched ACA negative groups were also compared.

### Statistical analysis

Normal distribution of continuous variables was assessed by the Kolmogorov-Smirnov test. Based on whether the variables were normally distributed or not, they were evaluated with either the Student's T-test or Mann-Whitney U test. Continuous variables were presented as mean ± SD or median with interquartile range (IQR). Binary variables were presented as a percentage, and the chi-squared test and Fisher's exact test were used. Inter- and intra-reader reliability were calculated by intraclass correlation coefficients (ICC). Linear regression analysis was performed to find the factors associated with SGUS score, and the factors with $P$ values under 0.1 in univariate analysis were included in multivariate analysis. $P$ values $< 0.05$ were considered statistically significant. All statistical tests were performed using the software R (R for Windows 3.3.2; The R Foundation for Statistical Computing, Vienna, Austria).

## Results

### Baseline characteristics of demographic, laboratory, and clinical findings

We performed SGUS on a total of 267 patients, but 146 of these patients were excluded from analysis. Finally, 19 pSS patients who were ACA positive (15.7%) and 102 who were ACA negative (84.3%) were incorporated in the analysis (Fig 1). The ACA-positive group was significantly older than the ACA-negative group at the time of diagnosis of pSS (58.2 ± 11.6 vs 49.9 ± 12.5 years, $P = 0.009$). For laboratory data, anti-Ro/SSA and anti-La/SSB antibodies were found more frequently in the ACA-negative group. Degrees of xerophthalmia measured by the Schirmer I test was comparable between the two groups, whereas USFR was lower in ACA-positive pSS group. Other characteristics are summarized in Table 1.

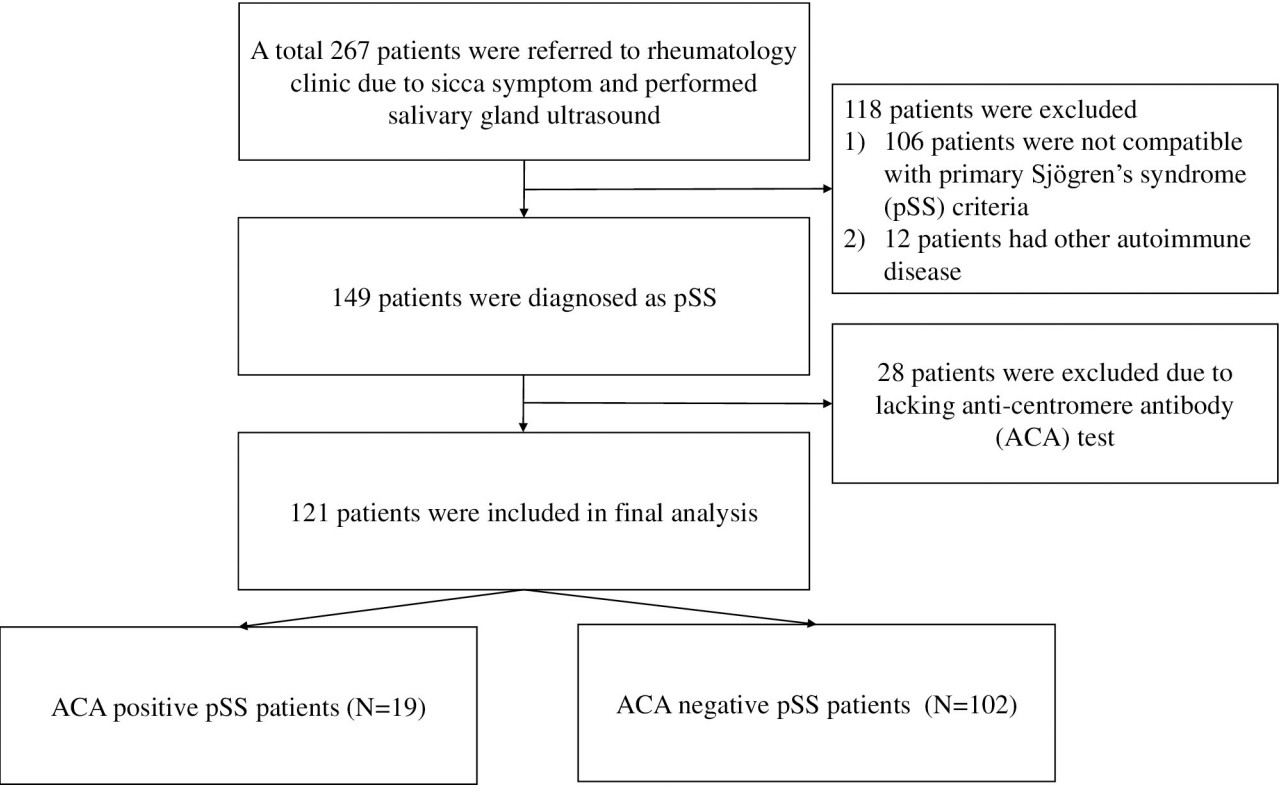

**Fig 1. Flow chart of included and excluded patients the in present study.**

## Comparison of SGUS scores between ACA-positive and ACA-negative pSS patients

The ICC of total SGUS score between reader 1 and 2 was 0.780 (95% CI 0.743–0.815), and ICCs of total SGUS score for intra-reader reliability were 0.821 (95% CI 0.778–0.853, reader 1) and 0.792 (95% CI 0.749–0.830, reader 2), respectively. The SGUS scores of the two groups were compared for total score, hypoechoic area score, hyperechoic foci score, and PDUS score. The hypoechoic area scores were significantly lower in the ACA-positive group than in the ACA-negative group (4.0 vs 7.0, $P = 0.004$), a difference which was also observed for total SGUS score (16.0 vs 23.0, $P = 0.027$). However, when comparing the number of patients with a total SGUS score $\geq$ 14, the outcomes were similar between the two groups. Detailed information of SGUS scores is presented in Table 2. In addition, PS-matched ACA-negative group (N = 38) was selected by imputing age and USFR in PS matching, then compared the SGUS score with ACA-positive group (N = 19). These also showed significant lower total SGUS score and SGUS score of hypoechoic area in ACA-positive group than PS-matched ACA-negative group (S1 Table).

## Predictors for total SGUS score

Following univariate linear regression analysis to identify factors associated with SGUS score, USFR positive (USFR $\leq$ 1.5 ml/15 min), anti-Ro/SSA positive, and RF positive results were significantly associated with SGUS score. Being positive for ACA showed a negative association with SGUS score ($\beta$ coefficient -8.51, $P = 0.039$). In multivariate analysis, USFR positive ($\beta$ coefficient 11.96, $P < 0.001$), anti-Ro/SSA positive ($\beta$ coefficient 5.97, $P = 0.010$), and RF

**Table 1. Baseline characteristics of patients with ACA positive and negative primary Sjogren's syndrome.**

| | Anti-centromere antibody positive pSS, (N = 19) | Anti-centromere antibody negative pSS, (N = 102) | P |
|---|---|---|---|
| Female, N (%) | 18 (94.7%) | 95 (93.1%) | 1.000 |
| Age (years) | 65.1 ± 8.2 | 54.2 ± 12.6 | 0.002 |
| Age at pSS diagnosis (years) | 58.2 ± 11.6 | 49.9 ± 12.8 | 0.009 |
| Disease duration (years) | 2.5 [0.0; 6.0] | 3.9 [0.2; 7.2] | 0.233 |
| Xerostomia, N (%) | 19 (100.0%) | 83 (81.4%) | 0.088 |
| Xerophthalmia, N (%) | 14 (73.7%) | 79 (77.5%) | 0.951 |
| Raynaud phenomenon, N (%) | 6 (31.6%) | 25 (24.5%) | 0.717 |
| ILD, N (%) | 2 (10.5%) | 8 (8.0%) | 1.000 |
| Schirmer's I test (mm/5min) | 3.0 [1.0; 5.0] | 3.0 [2.0; 5.0] | 0.846 |
| Schirmer's I test positive, N (%) | 12/14 (85.7%) | 49/65 (75.4%) | 0.628 |
| USFR (ml/15min) | 0.8 [0.2; 1.5] | 1.4 [1.0; 3.5] | 0.026 |
| USFR test positive, N (%) | 12/14 (85.7%) | 54/97 (55.7%) | 0.064 |
| ANA positive, N (%) | 15 (78.9%) | 54 (52.9%) | 0.064 |
| Anti-Ro antibody positive, N (%) | 4 (21.1%) | 91 (89.2%) | < 0.001 |
| Anti-La antibody positive, N (%) | 2 (10.5%) | 48 (47.1%) | 0.007 |
| RF positive, N (%) | 4 (21.1%) | 42 (42.0%) | 0.144 |
| IgG (mg/dL) | 1403.0 [1118.0;1681.5] | 1626.0 [1348.0;1942.0] | 0.053 |
| Hypergammaglobulinemia, N (%) | 7/15 (46.7%) | 51/93(54.8%) | 0.757 |
| IgA (mg/dL) | 272.0 [217.5;428.5] | 280.0 [222.5;389.0] | 0.910 |
| IgM (mg/dL) | 145.0 [98.0;258.5] | 96.0 [69.0;129.0] | 0.004 |
| C3 (mg/dL) | 103.7 [85.9;114.2] | 95.5 [89.3;107.8] | 0.535 |
| hypoC3, N (%) | 4/14 (28.6%) | 17/92 (18.5%) | 0.601 |
| C4 (mg/dL) | 25.8 [22.4;28.9] | 22.3 [19.1;26.2] | 0.097 |
| hypoC4, N (%) | 1/14 (7.1%) | 18/92 (19.6%) | 0.450 |
| ESR (mm/hr) | 14.0 [7.0;23.0] | 19.0 [10.0;30.0] | 0.159 |
| hs-CRP (mg/dL) | 0.1 [0.1; 0.1] | 0.1 [0.0; 0.1] | 0.614 |
| White blood cell (× 103/mm3) | 6290.0 [4645.0;7540.0] | 4945.0 [3940.0;6210.0] | 0.058 |
| Hemoglobin (g/dl) | 13.0 [12.2;13.7] | 12.6 [12.0;13.3] | 0.244 |
| Platelet (× $10^3$/mm$^3$) | 225.0 [167.0;252.0] | 218.0 [185.0;260.0] | 0.514 |

ANA, anti-nuclear antibody; C3, complement 3; C4, complement 4; ESR, erythrocyte sedimentation rate; hs-CRP, high-sensitivity C-reactive protein; Ig, immunoglobulin; ILD, interstitial lung disease; pSS, primary Sjögren's syndrome; RF, rheumatoid factor; USFR, unstimulated whole saliva flow rate.

Continuous variables are presented as mean ± standard deviation or median with interquartile range depending on whether it is normally distributed or not.

positive results (β coefficient 5.01, $P$ = 0.002) displayed significant association with SGUS score (Table 3).

## Discussion

The present study compared the SGUS findings between ACA-positive and ACA-negative pSS patients for the first time. The total SGUS score was significantly lower in the ACA-positive group, a difference that was more prominent in the hypoechoic area score than the hyperechoic foci score. Previous studies have shown that SGUS findings correlate with visual analogue scale of ocular and oral dryness, stimulated whole saliva flow rate, and USFR [11,12]. Also, higher SGUS score was associated with poor response to rituximab [22]. Among the SGUS scoring components, the extent of hyperechoic foci has shown significant positive association with USFR [13], and PDUS has shown predictive value for SGUS progression in 2-year follow up data [23]. In addition, SGUS of hypoechoic area significantly progressed in pSS

**Table 2. Salivary gland ultrasonography scores of patients with ACA positive and negative primary Sjogren's syndrome.**

| | Anti-centromere antibody positive pSS, (N = 19) | Anti-centromere antibody negative pSS, (N = 102) | P |
|---|---|---|---|
| Hypoechoic area score of parotid gland (0–6) | 2.0 [1.5; 2.5] | 4.0 [1.0; 4.0] | 0.011 |
| Hypoechoic area score of submandibular gland (0–6) | 2.0 [1.0; 3.0] | 4.0 [2.0; 4.0] | 0.019 |
| Total hypoehoic area score (0–12) | 4.0 [3.5; 5.0] | 7.0 [4.0; 9.0] | 0.004 |
| Hyperechoic foci score of parotid gland (0–6) | 2.0 [2.0; 2.0] | 2.0 [2.0; 3.0] | 0.860 |
| Hyperechoic foci score of submandibular gland (0–2) | 2.0 [2.0; 2.0] | 2.0 [2.0; 2.0] | 0.910 |
| Total hyperechoic foci score (0–8) | 4.0 [3.5; 4.0] | 4.0 [3.0; 4.0] | 0.789 |
| Echogenicity score of parotid gland (0–2) | 0.0 [0.0; 1.0] | 0.0 [0.0; 2.0] | 0.321 |
| Echogenicity score of submandibular gland (0–2) | 0.0 [0.0; 2.0] | 2.0 [0.0; 2.0] | 0.204 |
| Total echogenicity score (0–4) | 0.5 [0.0; 2.0] | 2.0 [0.0; 4.0] | 0.185 |
| Homogeneity score of parotid gland (0–6) | 2.0 [2.0; 4.0] | 2.0 [1.0; 4.0] | 0.775 |
| Homogeneity score of submandibular gland (0–6) | 3.5 [2.0; 5.0] | 4.0 [2.0; 6.0] | 0.754 |
| Total homogeneity score (0–12) | 6.0 [4.0; 8.0] | 6.5 [4.0;10.0] | 0.597 |
| Clearance of the border score of parotid gland (0–6) | 0.0 [0.0; 0.0] | 0.0 [0.0; 2.0] | 0.107 |
| Clearance of the border score of submandibular gland (0–6) | 2.0 [0.0; 3.0] | 2.0 [0.0; 3.0] | 0.865 |
| Total clearance of the border score (0–12) | 2.0 [0.0; 4.0] | 3.0 [0.0; 4.0] | 0.526 |
| PDUS score of parotid gland (0–6) | 1.5 [0.0; 3.0] | 2.0 [0.0; 3.0] | 0.611 |
| PDUS score of submandibular gland (0–6) | 1.0 [0.0; 4.0] | 2.0 [0.0; 3.0] | 0.993 |
| Total PDUS score (0–12) | 2.5 [0.0; 6.0] | 3.0 [0.0; 5.0] | 0.797 |
| SGUS score of parotid gland, (0–26) | 7.0 [4.0; 10.0] | 10.0 [4.0;14.0] | 0.097 |
| SGUS score of submandibular gland, (0–22) | 9.0 [5.5;12.0] | 13.0 [8.0;16.0] | 0.036 |
| Total SGUS score, (0–48) | 16.0 [11.5;21.5] | 23.0 [12.0;28.0] | 0.027 |
| SGUS score ≥ 14, N (%) | 12 (63.2%) | 72 (70.6%) | 0.708 |

PDUS, power Doppler ultrasonography; SGUS, salivary gland ultrasonography.

Continuous variables are presented as mean ± standard deviation or median with interquartile range depending on whether it is normally distributed or not.

patients [23]. Patients with pSS who had systemic manifestation (peripheral neuropathy or leucocytoclastic vasculitis or interstitial lung disease or lymphadenopathy or arthritis) had higher SGUS score [24]. Similar to past research, a positive USFR showed the strongest association with SGUS score in the present study. This suggests that SGUS score is closely related with actual salivary gland function. Although some studies demonstrated the association between specific component of SGUS and severity of sicca symptoms and predictive role of SGUS findings on prognosis of pSS, however, further studies are needed to clarify the role of SGUS findings in severity and prognosis of pSS patients. Several different SGUS scoring systems exist [9,20,25,26], the Outcome Measures in Rheumatology (OMERACT) ultrasound working group are developing a new scoring system, which may unify the current schemes and increase the sensitivity and specificity of the existing SGUS scoring system [27].

In the present study, SGUS score was also significantly related to being positive for anti-Ro/SSA and RF. A previous study showed that pSS patients who are anti-Ro/SSA positive, or double positive for both anti-Ro/SSA and anti-La/SSB, present a higher SGUS score than negative patients [24], and we have also reported that being double positive for anti-Ro/SSA and anti-La/SSB is positively associated with SGUS severity [21]. Zhang. X. et al demonstrated that SGUS score correlates with serum RF levels [25], which is similar to our current findings. Although the serologic levels of anti-Ro/SSA, anti-La/SSB, and RF have consistently shown a clear association with SGUS severity, the functional mechanisms of these antibodies in salivary

**Table 3. Associated factors with salivary gland ultrasonography score found by univariate and multivariate linear regression analysis.**

| | Univariate | | | Multivariate | | |
|---|---|---|---|---|---|---|
| | β | 95% CI | *P* | β | 95% CI | *P* |
| Age | 0.02 | -0.17, 0.22 | 0.799 | | | |
| Male gender | -6.62 | -15.53, 2.30 | 0.142 | | | |
| Disease duration (years) | 0.63 | -0.17, 1.44 | 0.121 | | | |
| Xerostomia | 5.14 | -1.78, 12.07 | 0.142 | | | |
| Xerophthalmia | 1.61 | -4.70, 7.91 | 0.612 | | | |
| Raynaud phenomenon | -0.16 | -6.30, 5.99 | 0.959 | | | |
| ILD | -6.83 | -15.00, 1.35 | 0.100 | | | |
| Schirmer I test positive | 4.05 | -2.18, 10.27 | 0.198 | | | |
| USFR positive | 9.91 | 5.34, 14.49 | <0.001 | 11.96 | 8.84, 15.07 | <0.001 |
| ANA positive | 3.67 | -1.48, 8.81 | 0.159 | | | |
| Anti-Ro/SSA positive | 6.91 | 0.88, 12.94 | 0.026 | 5.97 | 1.45, 10.49 | 0.010 |
| Anti-La/SSB positive | 1.84 | -3.45, 7.13 | 0.488 | | | |
| Anti-centromere positive | -8.51 | -16.57, -0.45 | 0.039 | -2.93 | -8.22, 2.36 | 0.275 |
| RF positive | 7.77 | 3.03, 12.50 | 0.002 | 5.01 | 1.84, 8.18 | 0.002 |
| Hypergammaglobulinemia | 2.58 | -2.59, 7.75 | 0.321 | | | |

Total $R^2$: 0.444, adjusted $R^2$: 0.423, $P<0.001$, β: Regression coefficient.

ANA, anti-nuclear antibody; ILD, interstitial lung disease; RF, rheumatoid factor; USFR, unstimulated salivary flow rate.

gland structural damage have not yet been revealed. Further research should be performed to elucidate the pathologic effects of anti-Ro/SSA, anti-La/SSB, and RF on the salivary glands.

Patients who are positive for ACA are perceived as a unique sub-group of pSS patients, and the prevalence is 3.7–27% [15]. Being positive for ACA is associated with an older age, a higher prevalence of Raynaud phenomenon, and a lower prevalence of hypergammaglobulinemia and leukopenia [17,28–33]. When analyzing autoimmune antibody profile, anti-Ro/SSA, anti-La/SSB, and RF show consistently lower positivity in ACA-positive pSS patients than in negative patients [29,31,32]. The results from the current study are consistent with those of previous research for several clinical and serologic features, such as older age and lower anti-Ro/SSA and anti-La/SSB prevalence in the ACA-positive pSS group. However, here the RF positivity, leukopenia, and hypergammaglobulinemia only showed a tendency for lower occurrence in ACA-positive patients, but this was not significant. These discordances may have risen from a relatively small sample size and a difference in race. Despite these small discrepancies with past research, we showed for the first time that the changes observed with SGUS were less severe in ACA-positive pSS patients than in negative patients.

Usually, ACA is present in limited-type systemic sclerosis patients [34], and can recognize several epitopes of centromere proteins; CENP-A, CENP-B, and CENP-C [35,36]. Minor salivary gland biopsy has revealed that the presence of fibrous tissue in the minor salivary glands of ACA-positive pSS patients is more severe [30]. Furthermore, exocrine gland dysfunction, measured by the Schirmer I test and USFR, is also enhanced in ACA-positive pSS groups when compared to negative groups [32]. Therefore, we expected the extent of hyperechoic foci, which potentially represent fibrous change in salivary glands, to be more severe in ACA-positive pSS patients. However, this parameter was similar between the ACA-positive and negative pSS groups in the present study. Further studies demonstrating correlation between each component of the SGUS scoring system and salivary gland biopsy observations could clarify these unexpected results.

Future research should also address several of the limitations of the current study. First, the sample size of the ACA-positive pSS group was relatively small. However, the prevalence of ACA positivity in the pSS group as a whole was 15.7%, which was comparable with previous studies dealing with the characteristics of ACA-positive pSS patients. Second, the study design was cross-sectional and no follow up data was evaluated. Third, we included pSS patients with various disease durations, not only newly diagnosed pSS patients. The thorough evaluation of SGUS, minor salivary gland biopsy, exocrine gland function (Schirmer I test and USFR), and laboratory data in pSS patients at the time of first diagnosis could clarify the difference between ACA-positive and negative patients. Finally, we collected the data retrospectively, and thus, some clinical information, such as the EULAR Sjögren's syndrome disease activity index, was lacking.

## Conclusions

In conclusion, we presented the difference between the SGUS findings of ACA-positive and negative pSS patients. The total SGUS score was less severe in ACA-positive pSS patients, and this difference was emphasized in the hypoechoic area component of the SGUS scoring system. This may indicate that ACA-positive pSS patients possess less severe exocrine gland damage than ACA-negative pSS patients.

## Supporting information

**S1 Table. Salivary gland ultrasonography scores of patients with ACA positive and age-, and USFR- propensity score matched ACA negative primary Sjogren's syndrome.** (DOCX)

## Author Contributions

**Conceptualization:** Hong Ki Min, Hae-Rim Kim.

**Data curation:** Hong Ki Min, Se-Hee Kim, Kyung-Ann Lee, Hae-Rim Kim.

**Formal analysis:** Hong Ki Min, Hae-Rim Kim.

**Investigation:** Hong Ki Min, Youngjae Park, Seung-Ki Kwok, Hae-Rim Kim.

**Methodology:** Hong Ki Min, Hae-Rim Kim.

**Resources:** Seung-Ki Kwok, Sang-Heon Lee.

**Writing – original draft:** Hong Ki Min, Hae-Rim Kim.

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
