## [Decision Letter · Decision Letter 0]

26 Jul 2021

PONE-D-21-19041

Ultrasonographic Characteristics of Major Salivary Glands in Anti-centromere Antibody-positive Primary Sjögren’s Syndrome

PLOS ONE

Dear Dr. Kim,

Thank you for submitting your manuscript to PLOS ONE. After careful consideration, we feel that it has merit but does not fully meet PLOS ONE’s publication criteria as it currently stands. Therefore, we invite you to submit a revised version of the manuscript that addresses the points raised during the review process.

Our reviewers found some interests in this study, but also pointed out a number of criticisms, which are relevant and critical. All comments made by reviewers must be adequately answered in the revised version.

We look forward to receiving your revised manuscript.

Kind regards,

Masataka Kuwana, MD, PhD

Academic Editor

PLOS ONE

“This research was supported by a grant from the Basic Science Research Program through the National Research Foundation of Korea (NRF) funded by the Ministry of Education, Science and Technology, Republic of Korea (NRF-2018R1D1A1A02050982, Receiver : HR Kim).”

5. Thank you for stating the following in the Funding Section of your manuscript:

“This research was supported by a grant from the Basic Science Research Program through the National Research Foundation of Korea (NRF) funded by the Ministry of Education, Science and Technology, Republic of Korea (NRF-2018R1D1A1A02050982).”

We note that you have provided funding information that is not currently declared in your Funding Statement. However, funding information should not appear in the Funding section or other areas of your manuscript. We will only publish funding information present in the Funding Statement section of the online submission form.

 “This research was supported by a grant from the Basic Science Research Program through the National Research Foundation of Korea (NRF) funded by the Ministry of Education, Science and Technology, Republic of Korea (NRF-2018R1D1A1A02050982, Receiver: HR Kim).”

Reviewers' comments:

Reviewer's Responses to Questions

**Comments to the Author**

1. Is the manuscript technically sound, and do the data support the conclusions?

Reviewer #1: Yes

Reviewer #2: No

Reviewer #3: Yes

2. Has the statistical analysis been performed appropriately and rigorously? 

Reviewer #1: Yes

Reviewer #2: Yes

Reviewer #3: Yes

3. Have the authors made all data underlying the findings in their manuscript fully available?

Reviewer #1: Yes

Reviewer #2: Yes

Reviewer #3: No

4. Is the manuscript presented in an intelligible fashion and written in standard English?

Reviewer #1: Yes

Reviewer #2: Yes

Reviewer #3: Yes

5. Review Comments to the Author

Reviewer #1: This is an interesting article on US characteristics in major salivary gland in patients with ACA+primary Sjoegren's syndrome. ACA+pSS is considered as a subgroup in SS, but differences in the pathogenesis are still unknown. In this study, authors found some characteristic US findings in ACA+pSS. As authors stated in limitation, number of patients in ACA+group is not many and it may be difficult to judge if these findings are definitive or not. ACA+SS is relatively few in all pSS and moreover may be heterogeneous in the various characteristics. US findings usually largely depends on the disease stage. I am not sure whether this kind of characteristics can be caused from existence of a certain type of antibody. If authors want to emphasize current conclusion, more definitive data would be required. Significance of US characteristics should be more discussed.

Reviewer #2: The authors investigated the salivary gland ultrasonography (SGUS) findings in primary Sjögren’s syndrome (pSS) patients positive for the anti-centromere antibody (ACA) and compared these with those in ACA-negative pSS patients. They conclude that ACA-positive pSS patients showed a lower SGUS score than ACA-negative patients, which was especially prominent in the hypoechoic area component. I think the lower SGUS score in the ACA-positive patients might be due to the significantly older age and lower USFR of this group. To show the possible contribution of ACA to the phenotype of SS, SGUS findings should be compared between age and severity matched groups.

What does the difference in hypoechoic area component mean? It should be discussed.

As ultrasound is frequently considered the most operator-dependent imaging technique, intra-rater and inter-rater reliability is an important concern. Results of intra- and inter-reader reliability studies should be reported.

Scanning procedure and ultrasound equipment should be written in more detail.

Reviewer #3: In this manuscript, the authors presented the difference of salivary gland ultrasonography (SGUS) findings between anti-centromere antibody (ACA)-positive and negative primary Sjögren’s syndrome (pSS) patients. The total SGUS score was less severe in ACA-positive pSS patients, and this difference was emphasized in the hypoechoic area component of the SGUS scoring system. In multivariate regression analysis, being positive for unstimulated salivary flow rate (USFR), anti-Ro/SSA, and rheumatoid factor (RF) showed significant association with the SGUS score. Although these findings might be clinically important and meaningful, the authors should address following points.

1) The authors described, ‘Degrees of xerophthalmia and xerostomia, measured by the Schirmer I test and USFR, were comparable between the two groups’. However, in Table1, USFR was significantly lower in ACA-positive pSS patients than in ACA-negative pSS patients.

2) In Table1, the pathological findings of labial salivary glands biopsy should be compared.

3) In Table2, other 3 components of SGUS score, such as echogenicity, homogeneity, and clearness of salivary gland borders should be added.

4) The reason or possible mechanism why hypoechoic area was decreased in ACA-positive pSS patients should be discussed.

6. PLOS authors have the option to publish the peer review history of their article (what does this mean?). If published, this will include your full peer review and any attached files.

Reviewer #1: No

Reviewer #2: No

Reviewer #3: No

---

## [Author Response · Author response to Decision Letter 0]

16 Sep 2021

Reviewer #1: This is an interesting article on US characteristics in major salivary gland in patients with ACA+primary Sjoegren's syndrome. ACA+pSS is considered as a subgroup in SS, but differences in the pathogenesis are still unknown. In this study, authors found some characteristic US findings in ACA+pSS. As authors stated in limitation, number of patients in ACA+group is not many and it may be difficult to judge if these findings are definitive or not. ACA+SS is relatively few in all pSS and moreover may be heterogeneous in the various characteristics. US findings usually largely depends on the disease stage. I am not sure whether this kind of characteristics can be caused from existence of a certain type of antibody. If authors want to emphasize current conclusion, more definitive data would be required. Significance of US characteristics should be more discussed.

Answer : We agree with the referee’s comment. In previous studies, ACA-positive pSS showed several differences with ACA-negative pSS (older age, high prevalence of Raynaud phenomenon, lower prevalence of hypergammaglobulinemia and leukopenia). In present study, some baseline characteristics (age, and USFR) differ between ACA-positive and ACA-negative groups. We selected age- and USFR- matched ACA-negative group, then compared SGUS findings of ACA-positive and age-/USFR- matched ACA-negative groups (supplementary Table 1). These were also similar with original SGUS comparison between ACA-positive and ACA-negative groups. We also added the significance of SGUS in discussion section (line 237-250). 

Reviewer #2: The authors investigated the salivary gland ultrasonography (SGUS) findings in primary Sjögren’s syndrome (pSS) patients positive for the anti-centromere antibody (ACA) and compared these with those in ACA-negative pSS patients. They conclude that ACA-positive pSS patients showed a lower SGUS score than ACA-negative patients, which was especially prominent in the hypoechoic area component. I think the lower SGUS score in the ACA-positive patients might be due to the significantly older age and lower USFR of this group. To show the possible contribution of ACA to the phenotype of SS, SGUS findings should be compared between age and severity matched groups.

Answer : As reviewer’s comment, we compared the SGUS score of ACA positive pSS with age- and USFR- matched ACA negative pSS group. The main results (lower SGUS score, lower hypoechoic area score of ACA positive pSS) were similar with original results. These were supplied as supplementary Table 1. 

What does the difference in hypoechoic area component mean? It should be discussed.

Answer : Although SGUS is useful tool for pSS diagnosis, however, little is known about the impact of hypoechoic area (SGUS score) in pSS prognosis. In one study, the hypoechoic area score significantly progressed in pSS patients ("Ultrasonographic Changes of Major Salivary Glands in Primary Sjögren's Syndrome." J Clin Med 2020; 9(3)). We added these on discussion section (line 242-243).

As ultrasound is frequently considered the most operator-dependent imaging technique, intra-rater and inter-rater reliability is an important concern. Results of intra- and inter-reader reliability studies should be reported.

Answer : We added intra- and inter-reader reliability on result section (line 198-200)

Scanning procedure and ultrasound equipment should be written in more detail.

Answer : We added detailed information of US scanning procedure and equipment (line 114-120, 142-145)

Reviewer #3: In this manuscript, the authors presented the difference of salivary gland ultrasonography (SGUS) findings between anti-centromere antibody (ACA)-positive and negative primary Sjögren’s syndrome (pSS) patients. The total SGUS score was less severe in ACA-positive pSS patients, and this difference was emphasized in the hypoechoic area component of the SGUS scoring system. In multivariate regression analysis, being positive for unstimulated salivary flow rate (USFR), anti-Ro/SSA, and rheumatoid factor (RF) showed significant association with the SGUS score. Although these findings might be clinically important and meaningful, the authors should address following points.

1) The authors described, ‘Degrees of xerophthalmia and xerostomia, measured by the Schirmer I test and USFR, were comparable between the two groups’. However, in Table1, USFR was significantly lower in ACA-positive pSS patients than in ACA-negative pSS patients.

Answer : We revised the sentence (line 182-184).

2) In Table1, the pathological findings of labial salivary glands biopsy should be compared.

Answer : We agree that minor salivary gland biopsy is important in pSS diagnosis. However, most of the enrolled pSS patients in present study did not performed minor salivary gland biopsy. As biopsy is invasive procedure, in our rheumatology clinics, only ambiguous patients (highly suspicious for pSS, but not fulfilled the classification criteria for pSS) were consulted for minor salivary gland biopsy.

3) In Table2, other 3 components of SGUS score, such as echogenicity, homogeneity, and clearness of salivary gland borders should be added.

Answer : We added information of echogenicity / homogeneity / clearness of salivary gland borders in Table 2. 

4) The reason or possible mechanism why hypoechoic area was decreased in ACA-positive pSS patients should be discussed.

Answer : Although SGUS is useful tool for pSS diagnosis, however, little is known about the role of hypoechoic area (SGUS score) in pSS. In addition, we originally expected that hyperechoic band may be increased in ACA-positive group, because ACA is usually found in systemic sclerosis which cause systemic fibrous change in various organs. We wrote this on discussion section (278-288). Suggesting possible mechanism at this stage may be cause overestimation of present study. 

We only found that the hypoechoic area score significantly progressed in pSS patients ("Ultrasonographic Changes of Major Salivary Glands in Primary Sjögren's Syndrome." J Clin Med 2020; 9(3)). We added these on discussion section (line 242-243).

---

## [Decision Letter · Decision Letter 1]

21 Oct 2021

Ultrasonographic Characteristics of Major Salivary Glands in Anti-centromere Antibody-positive Primary Sjögren’s Syndrome

PONE-D-21-19041R1

Dear Dr. Min,

We’re pleased to inform you that your manuscript has been judged scientifically suitable for publication and will be formally accepted for publication once it meets all outstanding technical requirements.

Kind regards,

Masataka Kuwana, MD, PhD

Academic Editor

PLOS ONE

Additional Editor Comments (optional):

Reviewers' comments:

Reviewer's Responses to Questions

**Comments to the Author**

1. If the authors have adequately addressed your comments raised in a previous round of review and you feel that this manuscript is now acceptable for publication, you may indicate that here to bypass the “Comments to the Author” section, enter your conflict of interest statement in the “Confidential to Editor” section, and submit your "Accept" recommendation.

Reviewer #1: All comments have been addressed

Reviewer #3: All comments have been addressed

2. Is the manuscript technically sound, and do the data support the conclusions?

Reviewer #1: Yes

Reviewer #3: Yes

3. Has the statistical analysis been performed appropriately and rigorously? 

Reviewer #1: Yes

Reviewer #3: Yes

4. Have the authors made all data underlying the findings in their manuscript fully available?

Reviewer #1: Yes

Reviewer #3: Yes

5. Is the manuscript presented in an intelligible fashion and written in standard English?

Reviewer #1: Yes

Reviewer #3: Yes

6. Review Comments to the Author

Reviewer #1: All comments have been addressed with new analysis and description. I do not have further comment on this article.

Reviewer #3: Authors have successfully revised the manuscript according to reviewers’ comments. I have no further comments.

7. PLOS authors have the option to publish the peer review history of their article (what does this mean?). If published, this will include your full peer review and any attached files.

Reviewer #1: No

Reviewer #3: No

---

## [Editor Report · Acceptance letter]

25 Oct 2021

PONE-D-21-19041R1 

Ultrasonographic Characteristics of Major Salivary Glands in Anti-centromere Antibody-positive Primary Sjögren’s Syndrome 

Dear Dr. Min:

I'm pleased to inform you that your manuscript has been deemed suitable for publication in PLOS ONE. Congratulations! Your manuscript is now with our production department. 

Kind regards, 

on behalf of

Prof. Masataka Kuwana 

Academic Editor

PLOS ONE